biomaterials/environmental science/materials science

$H_3PO_4$ pretreatment, bioadsorbent, palm kernel shell, coconut shell, optimization, adsorption

**Authors for correspondence:**
Paik San H'ng
e-mail: ngpaiksan@gmail.com
Kit Ling Chin
e-mail: kitling.chin419@gmail.com

This article has been edited by the Royal Society of Chemistry, including the commissioning, peer review process and editorial aspects up to the point of acceptance.

# Characterization of bioadsorbent produced using incorporated treatment of chemical and carbonization procedures

Chuan Li Lee[1], Paik San H'ng[1,2], Kit Ling Chin[1], Md Tahir Paridah[1,2], Umer Rashid[3] and Wen Ze Go[2]

[1]Institute of Tropical Forestry and Forest Products, [2]Faculty of Forestry, and [3]Institute of Advance Technology, Universiti Putra Malaysia, 43400 UPM Serdang, Selangor, Malaysia

CLL, 0000-0002-2293-4088

The production of bioadsorbent from palm kernel shell (PKS) and coconut shell (CS) pretreated with 30% phosphoric acid ($H_3PO_4$) was optimized using the response surface methodology (RSM). Iodine adsorption for both bioadsorbents was optimized by central composite design. Two parameters including the $H_3PO_4$ pretreatment temperature and carbonization temperature were determined as significant factors to improve the iodine adsorption of the bioadsorbent. Statistical analysis results divulge that both factors had significant effect on the iodine adsorption for the bioadsorbent. From the RSM analysis, it was suggested that using 80 and 79°C as $H_3PO_4$ pretreatment temperature and 714 and 715°C as carbonization temperature would enhance the iodine adsorption of the CS and PKS bioadsobent, respectively. These results indicated that $H_3PO_4$ is a good pretreatment for preparing PKS and CS prior to carbonization process to produce bioadsorbent with well-developed microporous and mesoporous volume. The effort to produce alternative high grade and inexpensive adsorbent derived from lignocellulosic biomass, particularly in the nut shell form was implied in this research.

## 1. Introduction

The expeditious industrialization and inadequate effluent treatment processes in various industries are creating a huge amount of low quality of water to water bodies. The existence of

heavy metals is one of the aspects that affect the quality of water [1]. The toxicity of heavy metal could create crucial problems such as the contamination of drinking water, the increase in atmosphere air concentrations near sources of emission, or ingestion through the food chain [2,3]. Additionally, the current environmental strategies to raise awareness have urged the research community to approach the economically feasible, development of robust and environmentally friendly processes capable of eliminating pollutants from water as well as to safeguard the health of affected populations [4]. Presently, a lot of technologies like adsorption, membrane filtration, precipitation and ion-exchange have been used to discard the metal pollutants from the aqueous solution [3,5]. Comparatively, adsorption process is an efficient way to expel the heavy metals, organic pollutants and inorganic matters from polluted waters [3,6–8]. The most commonly used adsorbent is coal-based carbon by virtue of its high surface area and high adsorption capacity [9].

Although using coal-based carbon adsorbent is an effective method to remove pollutants from water, the tremendous cost has provoked the exploration for alternatives and inexpensive adsorbents. Presently, the research to produce inexpensive adsorbents to remove the contaminants from wastewaters was increasing. Natural adsorbents or more fondly known as bioadsorbents were used in the removal of heavy metals [10]. It has been verified by various researchers [11–14] that lignocellulosic biomass wastes, especially nutshells, are the great contestants as precursors to remove the heavy metal ions from water. In Malaysia, enormous amounts of palm kernel shell (PKS) and coconut shell (CS) are produced as lignocellulosic biomass wastes. Chiefly, both of these lignocellulosic biomass wastes have the aptitude to be used as inexpensive adsorbents, as they not only serve as unused resources, but they are practically at hand and for the most part they are environmentally friendly [10,15]. Hence, the utilization of these inexpensive lignocellulosic biomass wastes as carbon precursor is very promising [12]. Consequently, lignocellulosic biomass PKS and CS were used as an alternative to expensive coal-based adsorbents in this study.

However, merely applying carbonization process on PKS and CS will only slightly enhance the carbon content and develop the initial porosity in the bioadsorbent, which resulted with porosity that is still comparatively low compared to coal. To address those problems, chemical pretreatment of PKS and CS by $H_3PO_4$ prior to carbonization was proposed in this study to obtain bioadsorbent with desired pore size distribution from inexpensive raw materials using low carbonization temperature and short time duration. The present work explores the possibility of incorporating chemical pretreatment and carbonization process to create bioadsorbents from CS and PKS with high adsorption value. To achieve substantial improvements in adsorption properties of bioadsorbent, a systematic approach to adsorption optimization is required. The sorption capacity of the bioadsorbent varied with precursor used as well as the time and temperature profile used in the carbonization process [16]. In this study, response surface methodology (RSM) was performed to evaluate the effect of $H_3PO_4$ pretreatment temperature and carbonization temperature on the iodine adsorption of bioadsorbent. RSM was also used to study the interaction effects of two parameters while iodine adsorption property was considered as responses. Additionally, the other desirable bioadsorbent properties including the fixed carbon content, ash content, volatile content, Brunauer–Emmett–Teller (BET) surface area and volume, Fourier transform infrared spectroscopy (FTIR) for surface chemistry studies, thermal behaviour through thermogravimetric analysis (TGA) as well as the structural analysis of the pore bodies and carbon crystals from the scanning electron microscope (SEM) were obtained to further evaluate the quality of the bioadsorbent produced. The derivation of the bioadsorbent from lignocellulosic biomass with high adsorption properties would alleviate problems of waste management. Meanwhile, it also provides a high-quality end product for wastewater treatment that could potentially expand the carbon market. Moreover, mechanisms responsible for the adsorption of the iodine onto the bioadsorbent will also be discussed in this paper.

# 2. Material and methods

## 2.1. Preparation of samples prior to $H_3PO_4$ pretreatment

Raw material CS was gleaned from the local stall located at Pasar Besar Kuantan, Pahang. The raw material PKS was gleaned from Seri Ulu Langat Palm Oil Mill, Dengkil Selangor. Both raw materials were cleaned and dried in an oven at 105°C for 48 h. After that, the dried raw materials were then crushed and sieved to 2–5 mm size range.

## 2.2. Integrated procedure of chemical pretreatment and carbonization

The specified mass of the dried PKS and CS were impregnated with 30% $H_3PO_4$ at an impregnation ratio of 1 : 1. The particles were, respectively, impregnated with $H_3PO_4$ at three different pretreatment temperatures; 70, 80 and 90°C for 2 h. After the chemical pretreatment process, the pretreated particles were filtered and washed with distilled water. The treated particles were dried at 105°C for 48 h. The dried samples were then carbonized with relative temperature 600, 700 and 800°C for 75 min. The detail of the standard method testing process and calculation for the characteristics of bioadsorbent included mass yield, fixed carbon content, ash content, volatile content, BET surface areas and surface characteristics were mentioned in the previous studies [17,18]. In addition, FTIR spectroscopy was carried out to analyse the surface functional groups of the non-treated and pretreated bioadsorbent. Spectra are recorded at a range of 700–4000 $cm^{-1}$. TGA is used to characterize the decomposition and thermal stability of materials under a variety of conditions and to examine the kinetics of the physico-chemical processes occurring in the sample [19]. TGA was done using Perkin Elmer USA equipment. The experiments were performed in an inert atmosphere with a continuous flow of argon at the rate of 50 ml $min^{-1}$ and heated at a heating rate of 10°C $min^{-1}$. About 10 mg of samples was placed on a balance located in the furnace tube and heat was applied over the temperature range from 0 to 1000°C.

## 2.3. Iodine adsorption

ASTM D4607-94 method was used to determine the iodine adsorption number for carbons. The iodine adsorption number was explained as the milligrams of iodine adsorbed by 1.0 g of carbon. A conical flask with 10 ml of 5% HCI and 1.0 g of activated carbon was swirled until the entire activated carbon was wetted. The wetted activated carbon was boiled for exactly 30 s and the solution was cooled to room temperature. To the mixture in the conical flask, 100 ml of 0.1 N (0.1 mol $l^{-1}$) iodine solution was then added. The mixture was later filtered using a Whatman 2 V filter paper. Finally, 50 ml of this filtrate was titrated with 0.1 N (0.1 mol $l^{-1}$) sodium thiosulphate in the presence of starch as indicator. The amount of iodine adsorbed per gram of carbon was calculated as follows:

$$\text{iodine adsorption} \left(\text{mg g}^{-1}\right) = \frac{\{(N_1 \times 126.93 \times N_2) - [(S_1 + H_1)/F_1] \times (S_1 \times 126.93) \times S_2\}}{M} \quad (2.1)$$

where $N_1$ is the iodine solution normality; $N_2$ the added volume of iodine solution; $H_1$ the added volume of 5% HCI (ml); $F_1$ the filtrate volume used in titration (ml); $S_1$ the sodium thiosulfate solution normality; $S_2$ the consumed volume of sodium thiosulfate solution (ml); $M$ the mass of carbon (g).

## 2.4. Statistical analysis

Statistical Package for the Social Science (SPSS) was used to analyse the data of physico-chemical properties of bioadsorbent for the analysis of variance (ANOVA) at 95% confident level ($p \leq 0.05$). The Tukey–Kramer multiple comparison test was used to determine the differences of the treatment effects when significance was observed. When the $p$-value was higher than 0.05 at the 95% confidence level, the effects were considered to be not statistically significant. The ANOVA for the mass yield, ash content, volatile content and fixed carbon content of CS and PKS bioadsorbent are presented in table 1.

The various process parameters for preparing the pretreated bioadsorbent were studied with RSM design called central composite design (CCD). This method is suitable for fitting a quadratic surface and it helps to optimize the effective parameters with a minimum number of experiments, and to analyse the interaction between the parameters. The design used for this study is face centred ($\alpha = 1$). The number of experimental runs from CCD for the two variables ($H_3PO_4$ pretreatment temperature and carbonization temperature) consists of 24 non-centre point runs and 5 centre point runs indicating that altogether 29 experiments were required. The two variables together with their respective ranges were chosen based on the literature and preliminary studies. $H_3PO_4$ pretreatment temperature and carbonization temperature are the vital parameters affecting the iodine adsorption of adsorbent [20]. For the complete design matrix of the experiments carried out, together with the results obtained, refer to table 2. The experimental sequence was randomized to minimize the effects of the uncontrolled factors. The response is the iodine adsorption of the bioadsorbent. An empirical model that correlated the response to the two preparation process variables using a second-degree polynomial equation was prepared.

**Table 1.** ANOVA for the mass yield, ash content, volatile content and fixed carbon content of CS and PKS bioadsorbent. Means followed by the same letter in the same column are not significantly different at $p \leq 0.05$ according to Tukey's multiple comparison test.

| pretreatment temperature (°C) | carbonized temperature (°C) | yield (%) | | ash content (%) | | volatile content (%) | | fixed carbon content (%) | |
|---|---|---|---|---|---|---|---|---|---|
| | | CS | PKS | CS | PKS | CS | PKS | CS | PKS |
| non-treated | 600 | 23.56 | 26.83 | 32.53 | 35.09 | 19.48 | 20.91 | 46.98 | 43.98 |
| non-treated | 700 | 20.95 | 24.57 | 35.44 | 38.54 | 16.11 | 17.31 | 48.44 | 44.15 |
| non-treated | 800 | 19.25 | 23.06 | 38.61 | 40.52 | 15.21 | 16.23 | 47.61 | 43.53 |
| 70 | 600 | $34.88_a$ | $34.80_a$ | $26.70_a$ | $26.07_a$ | $19.97_f$ | $23.77_f$ | $53.03_c$ | $49.88_c$ |
| 80 | 600 | $35.51_b$ | $34.39_b$ | $26.93_b$ | $27.04_b$ | $19.32_f$ | $22.27_e$ | $53.52_{bc}$ | $50.54_c$ |
| 90 | 600 | $33.50_c$ | $33.59_d$ | $29.17_e$ | $29.17_e$ | $17.54_e$ | $19.91_d$ | $53.10_c$ | $50.91_c$ |
| 70 | 700 | $37.75_a$ | $31.99_{bc}$ | $28.39_d$ | $27.38_d$ | $17.12_{de}$ | $19.16_d$ | $54.24_b$ | $53.15_b$ |
| 80 | 700 | $35.50_b$ | $31.08_e$ | $27.44_c$ | $27.26_c$ | $16.29_d$ | $17.55_c$ | $56.06_a$ | $55.09_a$ |
| 90 | 700 | $33.31_c$ | $30.92_f$ | $29.76_f$ | $29.31_f$ | $14.76_c$ | $17.52_c$ | $55.30_a$ | $53.16_b$ |
| 70 | 800 | $37.30_a$ | $29.20_c$ | $31.15_g$ | $30.62_g$ | $13.41_b$ | $16.33_c$ | $55.21_a$ | $52.76_b$ |
| 80 | 800 | $35.50_b$ | $29.13_e$ | $31.42_h$ | $30.76_h$ | $12.82_b$ | $14.15_b$ | $55.57_a$ | $55.00_a$ |
| 90 | 800 | $32.72_c$ | $27.85_g$ | $33.99_i$ | $36.01_i$ | $10.61_a$ | $10.83_a$ | $55.23_a$ | $53.15_b$ |
| p-value | | <0.001 | <0.001 | <0.001 | <0.001 | <0.001 | <0.001 | <0.001 | <0.001 |

**Table 2.** Experimental design matrix and results.

| run | factor 1 | factor 2 | response experiment | response predicted | response experiment | response predicted |
|---|---|---|---|---|---|---|
| | | | iodine adsorption | iodine adsorption | iodine adsorption | iodine adsorption |
| standard | A | B | CS bioadsorbent (mg g$^{-1}$) | | PKS bioadsorbent (mg g$^{-1}$) | |
| 1 | 70 | 600 | 333.83 | 324.49 | 331.03 | 323.48 |
| 2 | 70 | 600 | 333.83 | 324.49 | 331.03 | 323.48 |
| 3 | 70 | 600 | 333.83 | 324.49 | 331.03 | 323.48 |
| 4 | 90 | 600 | 322.66 | 316.27 | 314.28 | 311.38 |
| 5 | 90 | 600 | 325.45 | 316.27 | 319.86 | 311.38 |
| 6 | 90 | 600 | 322.66 | 316.27 | 319.86 | 311.38 |
| 7 | 70 | 800 | 350.58 | 351.64 | 347.79 | 347.99 |
| 8 | 70 | 800 | 353.37 | 351.64 | 350.58 | 347.99 |
| 9 | 70 | 800 | 350.58 | 351.64 | 347.79 | 347.99 |
| 10 | 90 | 800 | 314.28 | 320.15 | 314.28 | 315.42 |
| 11 | 90 | 800 | 319.86 | 320.15 | 317.08 | 315.42 |
| 12 | 90 | 800 | 319.86 | 320.15 | 314.28 | 315.42 |
| 13 | 70 | 700 | 358.96 | 367.23 | 353.37 | 360.72 |
| 14 | 70 | 700 | 358.96 | 367.23 | 350.58 | 360.72 |
| 15 | 70 | 700 | 356.17 | 367.23 | 353.37 | 360.72 |
| 16 | 90 | 700 | 342.20 | 347.37 | 331.03 | 338.38 |
| 17 | 90 | 700 | 342.20 | 347.37 | 333.83 | 338.38 |
| 18 | 90 | 700 | 342.20 | 347.37 | 331.03 | 338.38 |
| 19 | 80 | 600 | 406.43 | 424.01 | 375.71 | 388.95 |
| 20 | 80 | 600 | 406.43 | 424.01 | 372.92 | 388.95 |
| 21 | 80 | 600 | 409.22 | 424.01 | 375.71 | 388.95 |
| 22 | 80 | 800 | 442.73 | 439.52 | 403.64 | 403.23 |
| 23 | 80 | 800 | 439.94 | 439.52 | 400.84 | 403.23 |
| 24 | 80 | 800 | 442.73 | 439.52 | 403.64 | 403.23 |
| 25 | 80 | 700 | 470.66 | 460.92 | 431.56 | 421.07 |
| 26 | 80 | 700 | 467.86 | 460.92 | 428.77 | 421.07 |
| 27 | 80 | 700 | 470.66 | 460.92 | 428.77 | 421.07 |
| 28 | 80 | 700 | 467.86 | 460.92 | 428.77 | 421.07 |
| 29 | 80 | 700 | 470.66 | 460.92 | 431.56 | 421.07 |

# 3. Results and discussion

## 3.1. Physico-chemical characteristics of bioadsorbent from the integrated procedure

The physico-chemical properties of bioadsorbent derived from pretreated CS and PKS are summarized in table 1. For all the models, the ANOVA analysis reveals $p \leq 0.01$ for the parameters: pretreatment temperature, carbonization temperature and pretreatment temperature × carbonization temperature claims that the models were strongly significant at the 99% confidence level for CS and PKS bioadsorbent.

Table 1 summarizes the mass yield, ash content, volatile content and fixed content of pretreated bioadsorbent derived from CS and PKS after carbonization at different temperatures. Table 1 marks that $H_3PO_4$-preteated bioadsorbent gains the higher mass yield than the non-treated bioadsorbent. The yield

of adsorbent increased due to the presence of $H_3PO_4$ during pretreatment, which promotes depolymerization, dehydration and redistribution of constituent biopolymers, and also favours the conversion of aliphatic to aromatic compounds [21,22]. Nevertheless, as expected, the mass yield was reduced once the carbonization temperature was raised from 600 to 800°C. About 72% and 63% of the total weight of CS and PKS were reduced at 800°C, respectively. Such severe changes occur because the fruit shells were exposed to higher thermal environments when the temperature was increasing. The lignocellulose may break down into highly volatile gases and carbon char at elevated temperature. The disruption of the cellulose molecules can take place with the loss of CO and $CO_2$ and the formation of a structure with higher carbon assay, hence the significant weight loss during the carbonization process. Phosphoric acid acts as an acid catalyst and promotes bond cleavage reactions by cyclization and condensation during the activation process [23,24]. By reacting with organic species, it helps to cross-link biopolymer fragments via phosphate and polyphosphate bridges [23,25]. For each catalyst, the highest severity gave the lowest yield of hemicellulose as high severity pretreatment temperature tended to result in degradation of hemicellulose [26]. Therefore, the mass yield of $H_3PO_4$ pretreated bioadsorbent reduces when impregnation temperature increases from 70 to 90°C.

Low content of ash is an indicator of a high-quality carbon adsorbent. Low ash content typically leads to superior adsorption of organic compounds from aqueous solution due to the hydrophobicity of the material [27,28]. As reported by Chin *et al.* [29], lignocellulosic biomass without proper treatment will create operational problems related to ash effects during combustion. Throughout the mixing of lignocellulosic biomass with $H_3PO_4$ pretreatment, the acid restricts the formation of tar by the development of cross-links [30,31]. These may explain that the $H_3PO_4$ pretreated bioadsorbent achieved the lower total ash than the non-treated bioadsorbent in this study. Compared to the ash content of non-treated CS and PKS bioadsorbent carbonized at 700°C, $H_3PO_4$ pretreated bioadsorbents have 8.74–12.48% lower ash content. It is plausible that phosphorus from the acid may have integrated into the lignocellulosic biomass structure at higher temperatures, resulting in a higher ash content in the bioadsorbent [32–34].

Another effect of $H_3PO_4$ pretreatment in this study is the reduction in volatiles in the bioadsorbent. Volatiles and tars are undesirable when the desired product is char as they condense around the walls of the reactor and cause problems of clogging that necessitate frequent cleaning [23,35]. Following the thermal decomposition of the precursor, the chemical reacts with the product causing reduction in the evolution of volatile matter and inhibition of the particle shrinkage [36]. Thus, significant reduction in volatile content occurred when the lignocellulosic biomass was pretreated with $H_3PO_4$. Both lignocellulosic biomass pretreated with $H_3PO_4$ at 90°C and further carbonized at 800°C achieved the lowest value of volatile content. This may explain that the phosphoric acid functions to destroy the aliphatic and aromatic species present in lignocellulosic biomass and swiftly eliminate the volatile element during the process of carbonization [37]. Moreover, $H_3PO_4$ acts as a dehydrating agent that inhibits the formation of volatile substances during the carbonization process and will help to enhance the yield of porous carbon and to decrease the carbonization temperature and carbonization time [22,38–40]. Hence, bioadsorbent pretreated with $H_3PO_4$ acid achieved the lower volatile matter with low carbonization temperature.

Likewise, the proximate analysis indicates the higher fixed carbon content in $H_3PO_4$ pretreated bioadsorbent than in the non-treated bioadsorbent (table 1). The release of volatiles during carbonization will cause the elimination of the non-carbon species [39,40]. The fixed carbon content increases when the treatment temperature increases from 70 to 80°C and the carbonization temperature increases from 600 to 700°C, respectively. The increment of ash content and fixed carbon may account for the elimination of volatile matter in the sample during the carbonization process. Hence, the stable carbon as well as the ash-forming minerals were left behind [41]. Furthermore, the use of $H_3PO_4$ as chemical pretreatment agent promotes depolymerization, dehydration and redistribution of constituent biopolymer, thus increasing the fixed carbon content in activated carbon [21,42–44]. The highest carbon content for CS and PKS are 56.06 and 55.09% that was prepared via pretreatment with $H_3PO_4$ at 80°C and further carbonized at 700°C for 75 min as shown by the ANOVA analysis. The increase in carbon content is also related to the microporosity development [42]. High carbon content was aspired to achieve high BET surface area [45]. Aside from that, the adsorption capacity of the adsorbent is affiliated with the fixed carbon content and the specific surface area of the adsorbent [46].

## 3.2. Optimization of the integrated procedure on iodine adsorption value

The complete design matrix of the experiments carried out, together with the results obtained, are displayed in table 2. Data demonstrate that the experimental values obtained were in good agreement

**Table 3.** ANOVA for the model of iodine adsorption of pretreated bioadsorbent. *A*, pretreatment temperature; *B*, carbonized temperature.

| source | sum of squares | d.f. | mean square | F-value | p-value Prob > F |
|---|---|---|---|---|---|
| ANOVA for response surface quadratic model for CS bioadsorbent | | | | | |
| model | 87381.54 | 5 | 17 476.31 | 196.654 | <0.0001 |
| A | 1774.441 | 1 | 1774.441 | 19.96708 | 0.0002 |
| B | 1083.031 | 1 | 1083.031 | 12.18692 | 0.0020 |
| AB | 406.1385 | 1 | 406.1385 | 4.570117 | 0.0434 |
| $A^2$ | 72 235.46 | 1 | 72 235.46 | 812.8372 | <0.0001 |
| $B^2$ | 5719.178 | 1 | 5719.178 | 64.35565 | <0.0001 |
| residual | 2043.971 | 23 | 88.86831 | | |
| lack of fit | 1987.828 | 3 | 662.6093 | 236.0427 | <0.0001 |
| pure error | 56.14318 | 20 | 2.807159 | | |
| corrected total | 89 425.52 | 28 | | | |
| $R^2$ | 0.977143 | | | | |
| adjusted $R^2$ | 0.972174 | | | | |
| predicted $R^2$ | 0.964844 | | | | |
| adequate precision | 33.7336 | | | | |
| ANOVA for response surface quadratic model for PKS bioadsorbent | | | | | |
| model | 45620.5 | 5 | 9124.108655 | 124.433227 | <0.0001 |
| A | 2245.64 | 1 | 2245.643612 | 30.62575117 | <0.0001 |
| B | 916.764 | 1 | 916.7636395 | 12.50268518 | 0.0018 |
| AB | 314.452 | 1 | 314.4515701 | 4.288443405 | 0.0498 |
| $A^2$ | 34412.6 | 1 | 34 412.5575 | 469.3133039 | <0.0001 |
| $B^2$ | 4198.12 | 1 | 4198.12088 | 57.25334365 | <0.0001 |
| residual | 1686.48 | 23 | 73.32533985 | | |
| lack of fit | 1625.12 | 3 | 541.7056699 | 176.549678 | <0.0001 |
| pure error | 61.3658 | 20 | 3.06829033 | | |
| corrected total | 47307 | 28 | | | |
| $R^2$ | 0.964350268 | | | | |
| adjusted $R^2$ | 0.956600326 | | | | |
| predicted $R^2$ | 0.945816045 | | | | |
| adequate precision | 28.16196463 | | | | |

with the values predicted from the models with relatively small errors between the values. To fit the response function and experimental data, ANOVA, regression analysis and the interaction between the parameters were also studied and presented in table 3. The *F*-value estimated applying the experimental data correlated to the total residual and lack-of-fit values, respectively, and was lower than the tubular *F*-value. This marked that the iodine adsorption models were significant in this study scope. The iodine adsorption results obtained for all runs of the experimental design for the PKS and CS were represented by response surfaces. ANOVA results revealed pretreatment temperature and carbonization temperature to be very important factors as the *p*-value is less than 0.05. The strength of the effect of these two parameters on iodine adsorption is more distinct in the surface plot (figure 1*a*, *b*). The responses for the iodine adsorption number from the $H_3PO_4$ pretreatment and carbonization are depicted as two-dimensional surface plots of two factors: pretreatment temperature and carbonization temperature with their corresponding contour plots.

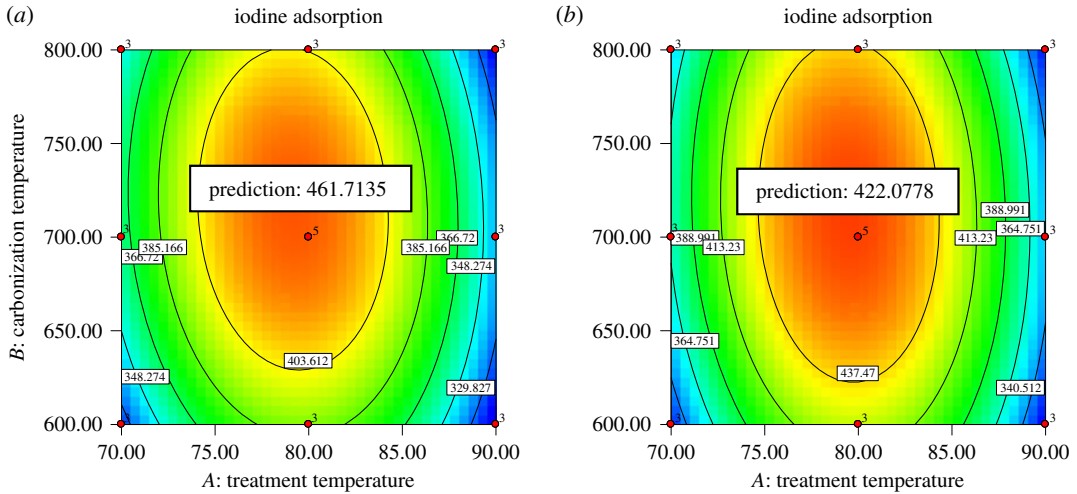

**Figure 1.** Response surface and contour plot of carbonization temperature versus treatment temperature: (*a*) CS bioadsorbent and (*b*) PKS bioadsorbent.

**Table 4.** The mathematical models derived from the experimental results for iodine adsorption. $T_1$, pretreatment temperature; $C_1$, carbonized temperature.

| bioadsorbent | model calculated for iodine adsorption (mg g$^{-1}$) |
|---|---|
| pretreated CS | $I_{cs} = -7900.31 + 168.8761(T_1) + 4.625001(C_1) - 0.00582(T_1)(C_1) - 1.03623(T_1)^2 - 0.00292(C_1)^2$ |
| pretreated PKS | $I_{pks} = -5627.7 + 116.901(T_1) + 3.97821(C_1) - 0.0051(T_1)(C_1) - 0.7152(T_1)^2 - 0.0025(C_1)^2$ |

The result marked that the pretreatment temperature and carbonization temperature had an equal influence on the iodine adsorption for CS and PKS bioadsorbent. Results declare that the $H_3PO_4$ pretreatment temperature significantly affected the iodine adsorption value of both bioadsorbents. It is evident that increasing the pretreatment temperature from 70 to 80°C serves to increase the iodine adsorption value of the bioadsorbent with a signification enhancement. However, as the $H_3PO_4$ pretreatment temperature increased from 80 to 90°C, the iodine adsorption value of the bioadsorbent decreased. This result attests that the maximum iodine adsorption value of both bioadsorbents was achieved when using 80°C as $H_3PO_4$ pretreatment temperature. Apart from that, iodine adsorption capability for both pretreated bioadsorbents was affected by the carbonization temperature, as displayed in figure 1. Figure 1*a,b* also illustrates the predicted peak in iodine adsorption number at the optimum condition. The optimum iodine adsorption was achieved when using 714 and 715°C as carbonization temperature for CS and PKS bioadsorbents, respectively.

Iodine adsorption value of the bioadsorbent was fitted to the model of the response surface provided by the mathematical model (table 4) to analyse the effect of $H_3PO_4$ pretreatment temperature and carbonization temperature on the iodine adsorption number. $I_{CS}$ and $I_{PKS}$ were the predicted iodine adsorption value (mg g$^{-1}$), where $T_1$ and $C_1$ denote pretreatment temperature and carbonization temperature. Numerical optimization using the response optimizer in Design-Expert v. 7.0.0 was carried out to find conditions for the maximized iodine adsorption number. The approach allowed to predict the optimal $H_3PO_4$ pretreatment temperature and carbonization temperature for a maximized iodine adsorption number. Table 5 shows the prediction condition design by RSM to achieve the optimum iodine adsorption value for both bioadsorbents. The predicted process parameters to gain the optimum value of iodine adsorption for CS bioadsorbent are 80°C as the pretreatment temperature and further carbonized at 714°C. On the other hand, to obtain the optimum iodine adsorption from PKS bioadsorbent the optimum $H_3PO_4$ pretreatment temperature and carbonization temperature are 79 and 715°C, respectively. The validity of these results were verified and confirmed through performing pretreatment temperature and carbonization temperature for each of the lignocellulosic biomass under the optimized conditions. For respective types of lignocellulosic biomass, the optimization experiment gave results of iodine adsorption as follows: 462.28 mg g$^{-1}$ for CS bioadsorbent and 423.18 mg g$^{-1}$ for PKS bioadsorbent. These experimental discoveries were in

**Table 5.** Predicted optimal pretreatment temperature and carbonization temperature with the corresponding iodine adsorption number.

| bioadsorbent | pretreatment temperature (℃) | carbonization temperature (℃) | predicted optimum (mg g$^{-1}$) | verification (mg g$^{-1}$) | residual of difference (%) |
|---|---|---|---|---|---|
| CS | 79.48 ≈ 80 | 713.81 ≈ 714 | 461.7135 | 462.2791 | 0.12 |
| PKS | 79.17 ≈ 79 | 715.10 ≈ 715 | 422.0778 | 423.1846 | 0.26 |

**Table 6.** Comparison of preparation and characteristics of bioadsorbent from this work with other studies.

| references | biomass | activation condition | iodine adsorption (mg g$^{-1}$) |
|---|---|---|---|
| present work | CS | $H_3PO_4$ pretreatment and activate with temperature of 714℃ | 462.28 |
| present work | PKS | $H_3PO_4$ pretreatment and activate with a temperature of 715℃ | 423.19 |
| [18] | CS | using temperature 700℃ with an activation time of 75 min | 348.74 |
| [18] | PKS | using temperature 700℃ with an activation time of 75 min | 304.97 |
| [47] | jute | activated with $N_2$ and steam with a temperature of 700℃ for 1 h | 338.00 |
| [48] | bamboo | using temperature 650℃ with an activation time of 30 min | 237.00 |

approximate agreement with the model prediction. The low residual value revealed the responses as 99.88% and 99.74% close to the predicted value for CS and PKS bioadsorbent, respectively. The validation experiment affirms that the predicted and experimental values are not significantly different, indicating that the CCD-RSM model developed is sufficient to describe the production of bioadsorbent with optimum iodine adsorption can be obtained with appropriate selections of $H_3PO_4$ pretreatment temperature and carbonization temperature. In the chemical pretreatment, the bioadsorbent obtained the highest iodine adsorption when the carbonization process was carried out at low temperature. Conclusively, bioadsorbent with highest iodine adsorption was formed by recombination reactions of $H_3PO_4$ pretreatment and carbonization process.

Iodine adsorption is the most fundamental parameter used to define and characterize the performance of adsorbent. As stated by Lee *et al.* [17], aside from affecting the development of the pores, $H_3PO_4$ pretreated bioadsorbent might have its unique structure which would considerably affect its adsorption behaviour. Table 6 shows the characteristics of bioadsorbents produced in this work and also from other recent studies by other researchers. It shows the results of the iodine adsorption of the bioadsorbent made from different precursors and condition of activation. Among them, the bioadsorbent produced in this study apparently possesses the best quality in terms of its adsorption capability. In other words, they have higher adsorption sites for molecules to attach onto the surface of the carbon. $H_3PO_4$ impregnation not only promotes the pyrolytic decomposition of raw material but also leads to the formation of cross-linked structure [17,49].

Evaluation of the surface characteristics and pore structures, surface chemistry (FTIR) and thermal stability (TGA) in this study was performed only on the bioadsorbent obtained using the optimal integrated procedure conditions; evaluation of the non-treated CS and PKS bioadsorbent that carbonized with temperature at 700℃ for 75 min and CS and PKS bioadsorbent pretreated with $H_3PO_4$ with temperature of 80℃ under carbonization temperature at 700℃ for 75 min.

## 3.3. Surface of the bioadsorbent

Elemental iodine can be bound to activated carbon by either chemisorption or physical adsorption. Physical adsorption is the initial method of adsorption on activated carbon, which is due to its large surface area and pore structure [50]. SEM micrographs (figure 2) are representing the surface of non-treated and $H_3PO_4$ pretreated bioadsorbents. We note that most of the pores are constricted and blocked in both non-treated bioadsorbents (figure 2a,b). Correspondingly, the surface morphology of

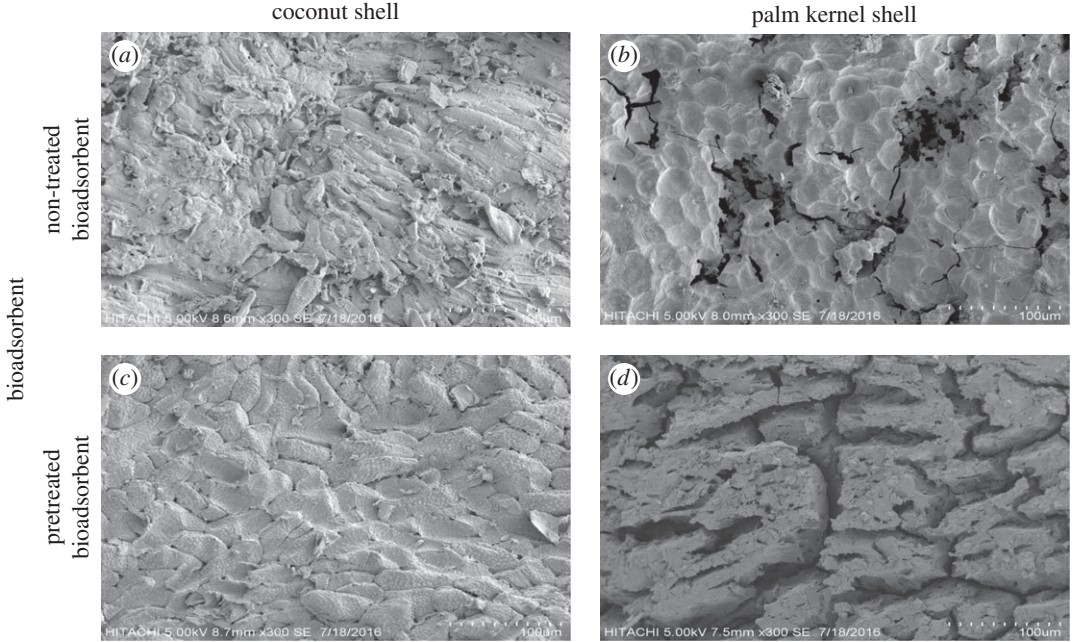

coconut shell palm kernel shell

**Figure 2.** Scanning electron micrograph of non-treated bioadsorbent CS (*a*) and PKS (*b*) carbonized with temperature at 700℃ for 75 min and bioadsorbent CS (*c*) and PKS (*d*) pretreated with $H_3PO_4$ with a temperature of 80℃ under carbonization temperature at 700℃ for 75 min.

both non-treated bioadsorbents shows cracks but no visible pore. The impregnation of precursor materials with chemical agents $H_3PO_4$ not only produce high yield of activated carbon products, it can inhibit tar formation and also reduce the volatile matter evolution which may result in better development of porous structure [51]. Thus, the phosphoric acid pretreatment preserved better starting structure of CS and PKS bioadsorbent. The cellular structure of CS and PKS was notable after chemical pretreatment and carbonization process. Large amounts of orderly pores are developing on the pretreated CS bioadsorbent surface with broken edges. This was attributed to the small amount of impurities such as tar that may cause the pore to clog up and inhibit good pore structure development [52]. At the same time, the formation of some cavities and rudimentary pores occurred on the surface of pretreated CS bioadsorbent. This may be clarified by the result of the space created by the volatilization of moisture content and the organic compounds from the carbon [53]. The SEM image of pretreated PKS bioadsorbent displays a barely systematized pore development or some rudimentary pores. This phenomenon was accounted to inadequate volatile release from the bioadsorbent. As reported from table 1, PKS bioadsorbent contains higher volatile content than CS bioadsorbent. The surface structures of the non-treated PKS have burnt out the pores with tunnel or honeycomb-like structures. The inconsistent pore structure and honeycomb structure on the surface of the adsorbent will lead to smaller surface area [54]. On the other hand, significant cracks were shown on the surface of $H_3PO_4$ pretreated PKS bioadsorbent. The thermal stress (carbonization) resulted in the development of cracks, crevices and slits in the matrix of the ultimate carbon material, as shown in figure 2*d*. Some researchers had also reported that the activation stage produced an extensive external surface (mesopore and macropore surface area) on the activated carbon with quite a number of irregular cracks and pores [54–56]. These pores result from the evaporation of the chemical reagent $H_3PO_4$ during carbonization leaving empty spaces [57,58].

## 3.4. BET surface area and pore size characterization

Table 7 illustrates the total BET surface area and pore size characterization of the prepared bioadsorbent. Chemical impregnation is one of the vital strides to produce higher quality bioadsorbent. Chemicals employed in this step are effective in decomposing the structure of the material as well as forming micropores, which may improve the adsorption of contaminants [59]. Li *et al.* [51] had mentioned that chemical activator inhibits the formation of volatile matter and other liquid that might deposit in the pore. When the chemical is removed by exhaustive washing, huge amount of porosity was formed [54,55]. With

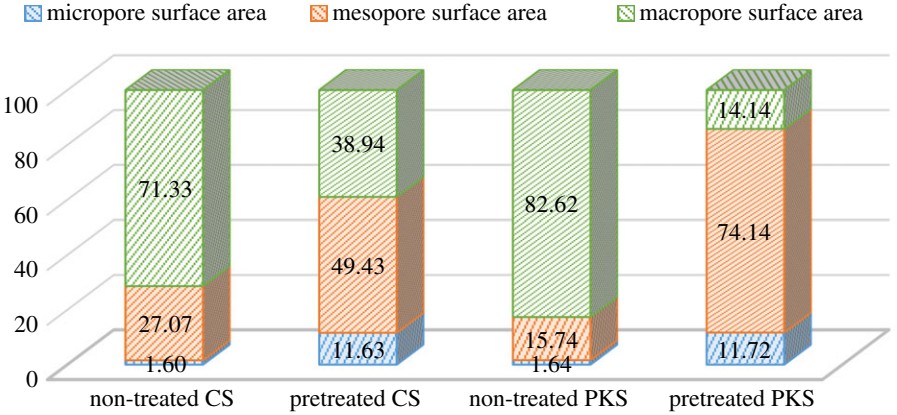

**Figure 3.** Ratio of porosity for the non-treated CS and PKS bioadsorbent carbonized with temperature at 700℃ for 75 min and CS and PKS bioadsorbent pretreated with $H_3PO_4$ with temperature of 80℃ under carbonization temperature at 700℃ for 75 min.

**Table 7.** The surface area and pore size characterization of the prepared bioadsorbent. BET surface area for the non-treated CS and PKS bioadsorbent carbonized with temperature at 700℃ for 75 min and CS and PKS bioadsorbent pretreated with $H_3PO_4$ with a temperature of 80℃ under carbonization temperature at 700℃ for 75 min.

| bioadsorbent | BET surface area ($m^2\,g^{-1}$) | pore volume ($cm^3\,g^{-1}$) | pore size (nm) |
| --- | --- | --- | --- |
| non-treated CS | 132.76 | 0.24 | 7.32 |
| pretreated CS | 204.64 | 0.28 | 5.52 |
| non-treated PKS | 101.92 | 0.20 | 7.66 |
| pretreated PKS | 135.97 | 0.24 | 7.05 |

reference to table 7, the highest BET specific surface area 204.64 $m^2\,g^{-1}$ was achieved by CS bioadsorbent treated with $H_3PO_4$ at 80°C and further carbonized with the temperature of 700°C. The high value of BET surface area in pretreated CS is indicative of highly developed pore network within the carbon. The result exposes that at a higher temperature, the reaction rate between char and the respective chemical activating agent led to the release of more volatile compounds generating more surface area and porosity. As stated by Lee *et al.* [18], lower carbonization temperature caused lesser volatile substances and tar to be released and produced underdeveloped carbon structures. On the contrary, the low surface areas for adsorbents are due to the absence of smaller pores like micro/mesopores [60]. According to table 7, pretreated bioadsorbent has smaller pore size than the non-treated bioadsorbent. The BET specific surface area of the $H_3PO_4$ pretreated PKS bioadsorbent (135.97 $m^2\,g^{-1}$) is higher than the non-treated PKS bioadsorbent. This phenomenon is in relation to the surface morphology of the pretreated PKS bioadsorbent (figure 2*d*). Activation stage ($H_3PO_4$ pretreatment) produces an extensive external surface with high surface area on the pretreated PKS bioadsorbent. Apart from that, the enlargement of surface area for both pretreated bioadsorbents may be accomplished via the creation of new micropores as propounded by the growth in the micropores' surface area and pore volume. $H_3PO_4$ is the dehydrating agent that penetrates deep into the structure of the carbon causing tiny pores to develop [17]. Simultaneously, it can be seen that the bioadsorbents derived from the CS and PKS by $H_3PO_4$ pretreatment under appropriate process conditions have higher proportions of micropores and mesopores with a high surface area. Figure 3 demonstrates that chemical agent has impacts on the macropore domain. The macropore ratio was highly reduced via $H_3PO_4$ pretreatment. Upon that, bioadsorbent with higher proportions of micro and mesoporosity but a fewer proportion of macroporosity can be produced via $H_3PO_4$ pretreatment. The molecular size of impurities in the purification of industrial wastewater is complex, and the presence of micropores and mesopores in carbon is favourable [61].

## 3.5. N$_2$ adsorption

Nitrogen adsorption is the standard channel to determine the porosity of carbonaceous adsorbent. The adsorption isotherm provided the information source for the porous structure of the adsorbent, heat

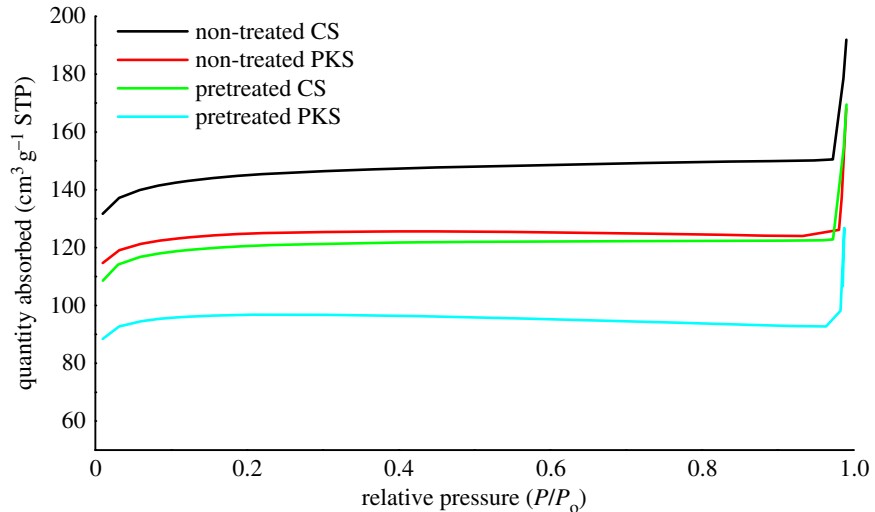

**Figure 4.** N$_2$-adsorption isotherm for the non-treated CS and PKS bioadsorbent carbonized with temperature at 700°C for 75 min and CS and PKS bioadsorbent pretreated with H$_3$PO$_4$ with temperature of 80°C under carbonization temperature at 700°C for 75 min at 77 K.

of adsorption, physical and chemical characteristics and others. Figure 4 establishes the N$_2$-adsorption isotherm acquired from the non-treated and H$_3$PO$_4$ pretreated bioadsorbent sample prepared from CS and PKS under the optimum conditions. Most isotherms have been shown to conform to one of the five types of IUPAC classification.

Both lignocellulosic biomasses pretreated with 30% of H$_3$PO$_4$ at a temperature of 80°C and further carbonized at a temperature of 700°C, respectively, were conforming to type IV isotherm. This discloses a significant microporous and mesoporous network system present in both the pretreated bioadsorbents. Their main differences depend on the experimental variables used and appear in their super-microporous and mesoporous range, presenting different isotherm slopes for relative pressures above 0.2 and up to unity (i.e. wider pore contributions) [62]. The conformation of type I isotherm is an indication that samples are given by microporous solids having relatively small external surfaces such as activated carbons, molecular sieve zeolites and certain porous oxides. Regarding figure 4, both pretreated bioadsorbents were found with materials having pore size distributions over a broader range including wider micropores and possibly narrow mesopores (less than −2.5 nm) [63]. Figure 4 signifies that the initial part of the type IV isotherm for carbon represents micropore filling, and the slope of the plateau at high relative pressure is due to multilayer adsorption on non-microporous surfaces, i.e. in mesopores, in macropores and on the external surface [64]. This infers that both H$_3$PO$_4$ pretreated bioadsorbents owned a mesoporous structure that contained a lot of mesopores and micropores and a few macropores. This proved that the non-toxic H$_3$PO$_4$ was the activating agent that leads to the development of micro and mesoporosity for bioadsorbent. Similar results demonstrated by Deshmukh *et al.* [60] revealed that H$_3$PO$_4$ pretreated samples mainly conform to type IV isotherm, typical of mesoporous materials with lower N$_2$ uptake at low relative pressure indicating the presence of some microporosity figure 5.

## 3.6. FTIR

The adsorption characteristics of activated carbon can be directly influenced by the material's surface chemistry. The acidity, polarity and presence of functional groups are dependent on the composition of the precursor and the production (carbonization/activation) process [65]. In this study, an attempt was made to evaluate the influence H$_3$PO$_4$ pretreatment for the preparation of bioadsorbent from CS and PKS. The functional and surface chemistry of the bioadsorbent was analysed using FTIR techniques. Spectra in wavenumbers 900–1300 cm$^{-1}$ are allocated specifically to phosphor species such as P=O, O–C stretching in P–O–C from aromatics and P=OOH [66]. Concurrently, the shoulder at 1180–1220 cm$^{-1}$ may be assigned to the stretching mode of hydrogen-bonded P=O, to O–C stretching vibrations in P–O–C (aromatic) linkage and to P=OOH [67]. FTIR result in this study denotes the strong peak at 1213 cm$^{-1}$ and 1212 cm$^{-1}$ on pretreated CS and PKS, respectively. When H$_3$PO$_4$ is

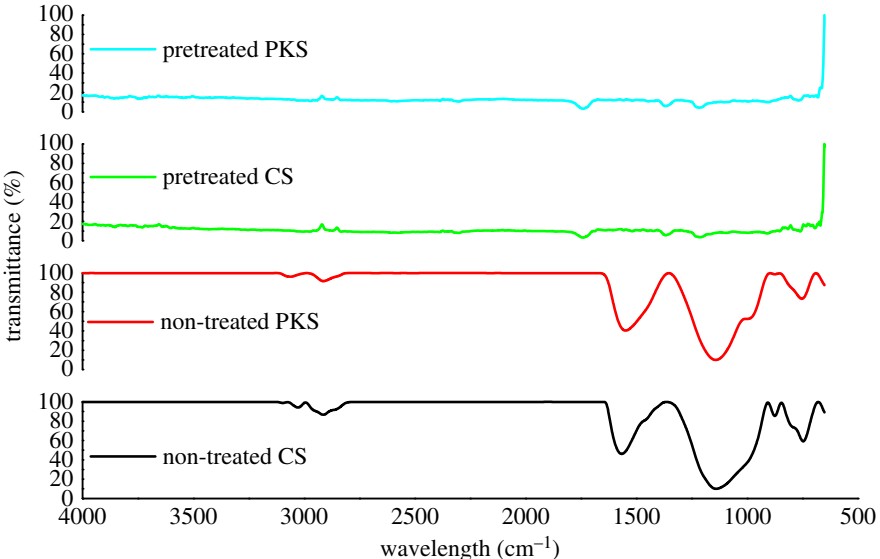

**Figure 5.** FTIR for the non-treated CS and PKS bioadsorbent carbonized with temperature at 700°C for 75 min and CS and PKS bioadsorbent pretreated with $H_3PO_4$ with temperature of 80°C under carbonization temperature at 700°C for 75 min.

mixed with lignocellulosic biomass at high temperature, it appears to function both as an acid catalyst to promote bond cleavage reactions and formation of cross-links, via processes such as cyclization and condensation, and to combine with organic species to form phosphate and polyphosphate bridges that connect and cross-link biopolymer fragments [25,41]. Contemporaneously, $H_3PO_4$ is also able to combine with organic species to form phosphate linkage, such as phosphate and polyphosphate ester [68].

$H_3PO_4$ pretreatment promotes two new sharp peaks at 1368 cm$^{-1}$ for CS and PKS bioadsorbent. The increasing bands at 1368–1370 cm$^{-1}$ have been reported to stem from bonds of lignin carbohydrate complexes, which were the main –CH$_3$ (lignin) and –CH$_2$ (carbohydrates) groups in CS and PKS bioadsorbent [69]. Strong stretching vibrations at 1740 cm$^{-1}$ that occur on the pretreated bioadsorbent may correspond to the presence of N–H bonds of the amines and O–H bonds of the carboxylic acids. The surface functional groups such as N–H and C–H stretching played a significant role in heavy metals adsorption [70]. Adsorption on amine-grafted materials may be a potentially attractive alternative to capture $CO_2$ from power plants. Activated carbon has been proposed as a potential adsorbent due to its natural affinity for $CO_2$ and the possibility of tailoring its textural properties and surface chemistry to enhance capacity and selectivity [67,68]. The difference between non-treated bioadsorbent and pretreated bioadsorbent was a strong peak located at 2302 and 2301 cm$^{-1}$. This peak connotes the presence of C≡C stretching vibrations in alkyne groups [68,71]. In addition, pretreated CS and PKS bioadsorbent manifests a weak but visible absorption at 2605 cm$^{-1}$ and 2602 cm$^{-1}$, respectively. This vibration was considered as S–H bond stretching vibration [72]. The FTIR result clearly demonstrates that –SH group had been successfully grafted onto the surface of pretreated bioadsorbent. For the purpose of increasing the adsorption efficiency, partial or full surface modification was required. Thiol-functionalization is one of the most important methods for cellulose modification. For example, thiosemicarbazide with C=S and amino groups have been grafted onto the surface of cellulose for heavy metal removal from aqueous solutions [73,74]. We may also note the $H_3PO_4$ pretreated bioadsorbent contains less C–H bond when compared with non-treated bioadsorbent. The chemical agent $H_3PO_4$ in activation process may dehydrate the residual organic molecules that prevent hydrocarbon deposition on the carbon surface, while the carbonization process at high temperature removes the absorbed hydrocarbon [51]. Cundari *et al.* [75] had prompted that weakening the C–H bond of the hydrocarbon yields a more strongly bound adduct.

## 3.7. TGA

By analysing both TGA results, it was observed that the $H_3PO_4$ pretreated bioadsorbent has very high resistance to weight loss. Three zones of the profile were obtained for both cases. Apparently by increasing temperature from 400 to 800°C, the total weight losses for pretreated CS and PKS are

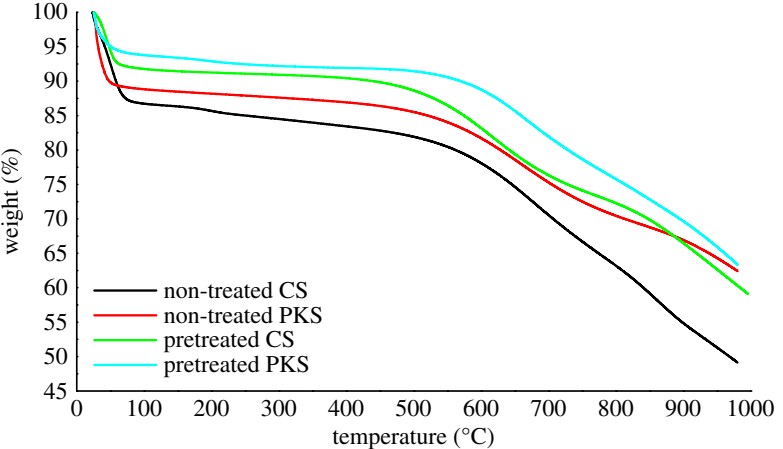

**Figure 6.** TGA for the non-treated CS and PKS bioadsorbent carbonized with temperature at 700°C for 75 min and CS and PKS bioadsorbent pretreated with $H_3PO_4$ with a temperature of 80°C under carbonization temperature at 700°C for 75 min.

17.68% and 15.51%, respectively. From this result (figure 6), it was found that both $H_3PO_4$ pretreated bioadsorbent exhibits high thermal stability. The presence of $H_3PO_4$ in the interior of the precursor restricts the tar formation of cross-links and inhibits the shrinkage of the precursor particle by occupying certain substantial volumes resulting in the lower weight loss and higher yield for $H_3PO_4$ impregnated carbon [71,72]. Complementarily, the contributions from alcoholic, C–O–C and C–O–P groups via $H_3PO_4$ pretreatment increased the thermal stability in the CS and PKS bioadsorbent. Some researchers had observed that the phosphate-like structure bound to carbon is the most abundant and thermally stable P species in phosphoric acid activated carbons over a wide range of temperature [23,76]. TGA result displays the weight per cent values for the residues of the $H_3PO_4$ pretreated bioadsorbent. Pretreated bioadsorbent resulted in a high amount of residue. Approximately, 59.10% and 63.37% of the mass are not volatilized even under 1000°C for CS and PKS bioadsorbent, respectively. $H_3PO_4$ is a dehydrating compound which will reduce the mass loss and change the thermal degradation of the precursor, leading to a subsequent change in the evolution of porosity [77]. Hence, it can be seen that the $H_3PO_4$ pretreated bioadsorbents have lower mass loss than non-pretreated bioadsorbents.

## 4. Conclusion

Results indicate that $H_3PO_4$ is a suitable chemical agent for the preparation of microporous and mesoporous bioadsorbent from CS and PKS under appropriate process condition. The RSM analysis further reveals that the effect of $H_3PO_4$ pretreatment temperature and carbonization temperature has a comparatively strong impact on the iodine adsorption of manufactured bioadsorbent. The optimum conditions for the preparation of the CS bioadsorbent are $H_3PO_4$ pretreatment temperature of 80°C and carbonization temperature of 714°C. However, the optimum conditions for the preparation of the PKS bioadsorbent are 79°C for $H_3PO_4$ prereatment temperature and carbonization temperature of 715°C. By using the optimum conditions, bioadsorbents with large BET specific surface area (high proportion of mesoporous and microporous) and highly thermally stable (with more functional group and lower mass loss) were produced. The results of this work provide insights into designing bioadsorbent with high adsorption capability by chemical pretreatment for specialized applications. Bioadsorbent is widely used in wastewater treatment for adsorption of organic substances and non-polar adsorbates. It is the most widely used adsorbent as most of its chemical and physical properties can be tuned during the production according to its usage and preference. Moreover, the bioadsorbents derived from CS and PKS were detected to have high specific surface area due to its high porous surface and high adsorption capacity. These results show that $H_3PO_4$ pretreated bioadsorbent is a promising catalyst support material.

Data accessibility. Data available from the Dryad Digital Repository at: https://doi.org/10.5061/dryad.mg5h6dr [78].
Authors' contributions. C.L.L.: data collection, analysis and interpretation of data, drafting the article or revising it critically for important intellectual content. P.S.H.: agreement to be accountable for all aspects of the work in ensuring that questions related to the accuracy or integrity of any part of the work are appropriately investigated and resolved.

M.T.P.: final approval of the version to be published. K.L.C.: drafted the article or revised it critically for important intellectual content. U.R.: final approval of the version to be published. W.Z.G.: analysis and interpretation of data.

Competing interests. We have no competing interests.

Funding. The authors are grateful for the financial support from co-author P.S.H. under the Higher Institution Centre of Excellence (HICoE) project at the Institute of Tropical Forestry and Forest Products which given by the Ministry of Higher Education Malaysia (MOHE).

Acknowledgements. The authors are grateful for the financial support given by the Ministry of Higher Education Malaysia (MOHE) under the Higher Institution Centre of Excellence (HICoE) project at the Institute of Tropical Forestry and Forest Products. The authors also sincerely thank the postgraduate students who participated in the field sampling exercise.

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
