## [Reviewer comments · Royal Society Open Science]

Review History

RSOS-190667.R0 (Original submission)

Review form: Reviewer 1 (Bogdan Marian I Tofanica)

Is the manuscript scientifically sound in its present form?

Yes

Are the interpretations and conclusions justified by the results?

Yes

Is the language acceptable?

Yes

Is it clear how to access all supporting data?

No

Do you have any ethical concerns with this paper?

No

Have you any concerns about statistical analyses in this paper?

No

Recommendation?

Reject

Comments to the Author(s)

Thank for the opportunity to review the article "Characterization of bioadsorbent produced using incorporated treatment of chemical and carbonization procedures" which addresses a fashionable research topic, relevant and important.

I appreciate the experimental work in the paper, but I do not feel that this research throw new light that deserve publication in a journal devoted to the development of scientific knowledge, because:

- the originality and the novelty of the manuscript: the main objective of the manuscript has already been developed and reported in other research articles (DOI: 10.1016/j.jclepro.2014.06.093, 10.1016/S0961-9534(96)00034-7, 10.1007/BF03326184, 10.4103/2423-7752.181802, BioResources 2016 - 11 (2) - 4485, 10.1016/j.proeng.2016.06.463, etc) because the subject is well known;

- the validity of the work: experimental and mathematical methods used are clearly explained and are appropriate to the experiments, but the results obtained don't surprise anybody, they fit in what is already known in the literature.

In conclusion, the novelty, the methodology and the results of the manuscript are quite common. The merits for publication are limited; therefore, the manuscript cannot meet the high quality of RSOS, a top-tier journal, dedicated to original research in science.

BM Tofanica

<https://publons.com/researcher/426293/bm-tofanica/>

Review form: Reviewer 2

Is the manuscript scientifically sound in its present form?

Yes

Are the interpretations and conclusions justified by the results?

Yes

Is the language acceptable?

Yes

Is it clear how to access all supporting data?

Yes

Do you have any ethical concerns with this paper?

No

Have you any concerns about statistical analyses in this paper?

No

Recommendation?

Major revision is needed (please make suggestions in comments)

Comments to the Author(s)

The manuscript entitled "Characterization of bioadsorbent produced using incorporated treatment of chemical and carbonization procedures." by Chuan Li Lee et. al. describes a method for produce an inexpensive adsorbent derived from lignocellulosic biomass. This article is interesting, but there are some questions need to be solved before it can be accepted for publication. Here is the detail of necessary revision,

1. There are some mistakes in the format of the manuscript. For example, the abstract is missing and the author needs to check the format of the first paragraph of all chapters.
2. The author needs to modify the format of this manuscript according to the requirements of journals.
3. It seems that all the figures in this manuscript are drawn with "Excel". "origin" is a better choice. It is suggested to modify all figures in this manuscript.
4. Please modify the unit in Table 6.
5. Please provide more advantages about this material for its application prospect.

Review form: Reviewer 3

Is the manuscript scientifically sound in its present form?

Yes

Are the interpretations and conclusions justified by the results?

Yes

Is the language acceptable?

Yes

Is it clear how to access all supporting data?

Yes

Do you have any ethical concerns with this paper?

No

Have you any concerns about statistical analyses in this paper?

No

Recommendation?

Major revision is needed (please make suggestions in comments)

Comments to the Author(s)

Comments to RSOS-190667:

Herein the author reported two novel iodine adsorbents derived from palm kernel and coconut shell and pretreated with phosphoric acid. H_3PO_4 prior to carbonization played significant role to develop porosity and further enhance the adsorbing ability. On the whole, this article is well organized, technically good, and has new ideas as well as good data support. Therefore, I recommend it for publication in the journal, but for better, it may need major revision:

1. How did the adsorption experiment conduct? Please describe it.

2. To understand the adsorption process, please provide reasonable and detailed adsorbing mechanism.
3. How does your work compete with the existing ones towards iodine adsorption? Please provide a comparative and comprehensive explanation.

Decision letter (RSOS-190667.R0)

25-Jun-2019

Dear Miss LEE:

Title: Characterization of bioadsorbent produced using incorporated treatment of chemical and carbonization procedures
Manuscript ID: RSOS-190667

The editor assigned to your manuscript has now received comments from reviewers. We would like you to revise your paper in accordance with the referee and Subject Editor suggestions which can be found below (not including confidential reports to the Editor). Please note this decision does not guarantee eventual acceptance.

Please submit your revised paper before 18-Jul-2019. Please note that the revision deadline will expire at 00.00am on this date. If we do not hear from you within this time then it will be assumed that the paper has been withdrawn. In exceptional circumstances, extensions may be possible if agreed with the Editorial Office in advance. We do not allow multiple rounds of revision so we urge you to make every effort to fully address all of the comments at this stage. If deemed necessary by the Editors, your manuscript will be sent back to one or more of the original reviewers for assessment. If the original reviewers are not available we may invite new reviewers.

RSC Associate Editor:

Comments to the Author:

Although Reviewer 1 did not support the publication of the manuscript, because Reviewer 2 and Reviewer 3 (Adjudicator) felt that the work would be suitable for publication after revisions, I am pleased to invite you to submit a revised manuscript.

RSC Subject Editor:

Comments to the Author:

(There are no comments.)

Reviewers' Comments to Author:

Reviewer: 1

Comments to the Author(s)

Thank for the opportunity to review the article "Characterization of bioadsorbent produced using incorporated treatment of chemical and carbonization procedures" which addresses a fashionable research topic, relevant and important.

I appreciate the experimental work in the paper, but I do not feel that this research throw new light that deserve publication in a journal devoted to the development of scientific knowledge, because:

- the originality and the novelty of the manuscript: the main objective of the manuscript has already been developed and reported in other research articles (DOI: 10.1016/j.jclepro.2014.06.093, 10.1016/S0961-9534(96)00034-7, 10.1007/BF03326184, 10.4103/2423-7752.181802, BioResources 2016 - 11 (2) - 4485, 10.1016/j.proeng.2016.06.463, etc) because the subject is well known;

- the validity of the work: experimental and mathematical methods used are clearly explained and are appropriate to the experiments, but the results obtained don't surprise anybody, they fit in what is already known in the literature.

In conclusion, the novelty, the methodology and the results of the manuscript are quite common. The merits for publication are limited; therefore, the manuscript cannot meet the high quality of RSOS, a top-tier journal, dedicated to original research in science.

BM Tofanica

<https://publons.com/researcher/426293/bm-tofanica/>

Reviewer: 2

Comments to the Author(s)

The manuscript entitled "Characterization of bioadsorbent produced using incorporated treatment of chemical and carbonization procedures." by Chuan Li Lee et. al. describes a method for produce an inexpensive adsorbent derived from lignocellulosic biomass. This article is interesting, but there are some questions need to be solved before it can be accepted for publication. Here is the detail of necessary revision,

1. There are some mistakes in the format of the manuscript. For example, the abstract is missing and the author needs to check the format of the first paragraph of all chapters.
2. The author needs to modify the format of this manuscript according to the requirements of journals.
3. It seems that all the figures in this manuscript are drawn with "Excel". "origin" is a better choice. It is suggested to modify all figures in this manuscript.
4. Please modify the unit in Table 6.
5. Please provide more advantages about this material for its application prospect.

Reviewer: 3

Comments to the Author(s)

Comments to RSOS-190667:

Herein the author reported two novel iodine adsorbents derived from palm kernel and coconut shell and pretreated with phosphoric acid. H_3PO_4 prior to carbonization played significant role to develop porosity and further enhance the adsorbing ability. On the whole, this article is well organized, technically good, and has new ideas as well as good data support. Therefore, I recommend it for publication in the journal, but for better, it may need major revision:

1. How did the adsorption experiment conduct? Please describe it.
2. To understand the adsorption process, please provide reasonable and detailed adsorbing mechanism.
3. How does your work compete with the existing ones towards iodine adsorption? Please provide a comparative and comprehensive explanation.

Author's Response to Decision Letter for (RSOS-190667.R0)

See Appendix A.

RSOS-190667.R1 (Revision)

Review form: Reviewer 3

Is the manuscript scientifically sound in its present form?

Yes

Are the interpretations and conclusions justified by the results?

Yes

Is the language acceptable?

Yes

Do you have any ethical concerns with this paper?

No

Have you any concerns about statistical analyses in this paper?

No

Recommendation?

Accept as is

Comments to the Author(s)

Accept

Decision letter (RSOS-190667.R1)

12-Aug-2019

Dear Miss LEE:

Title: Characterization of bioadsorbent produced using incorporated treatment of chemical and carbonization procedures

Manuscript ID: RSOS-190667.R1

It is a pleasure to accept your manuscript in its current form for publication in Royal Society Open Science. The chemistry content of Royal Society Open Science is published in collaboration with the Royal Society of Chemistry.

RSC Associate Editor:
Comments to the Author:
(There are no comments.)

RSC Subject Editor:
Comments to the Author:
(There are no comments.)

Reviewer(s)' Comments to Author:
Reviewer: 3

Comments to the Author(s)
Accept

Appendix A

Response to Reviewer's Comments Manuscript No. RSOS-190667

We are thankful to the reviewers for their thoughtful reviews and valuable comments on our manuscript. These comments are very constructive, and will help us to improve the manuscript, specifically in terms of clarifying our discussion and the goal of this paper. We take concerns seriously and have addressed them to the best of our abilities. Changes have been made as suggested by the reviewer. Some of the more notable changes are listed as below;

Reviewer 1:

Comments to the Author(s)

1. Experimental and mathematical methods used are clearly explained and are appropriate to the experiments, but the results obtained don't surprise anybody, they fit in what is already known in the literature. The novelty, the methodology and the results of the manuscript are quite common.

Response: Table 6 has been added in the manuscript (Table 6, p.8) to highlight the efficiency of the present bioadsorbent as compared to the other biomass-derived carbon materials. The comparison was made on the iodine adsorption properties of the bioadsorbents from different carbonization conditions under various treatment temperature and time. Among them, the bioadsorbent produced in this study apparently possesses the best quality in terms of its adsorption capability. In other words, they have higher adsorption sites for molecules to attach onto the surface of the carbon. The detailed discussion and reasons were added and clearly marked in p.8. References were added to provide a clear understanding for readers.

In this study, the RSM analysis further reveals the effect of H_3PO_4 pretreatment temperature and carbonization temperature has a comparatively strong impact on the iodine adsorption of manufactured bioadsorbent. No similar work has been reported particularly on the effect of impregnation temperature of H_3PO_4 on the bioadsorbent has suggests a need of the present work. Moreover, chemical pretreatment of PKS and CS by H_3PO_4 prior to carbonization has successfully produce bioadsorbent with desired pore size distribution from inexpensive raw materials using low carbonization temperature and short duration time.

2. The main objective of the manuscript has already been developed and reported in other research articles (DOI: 10.1016/j.jclepro.2014.06.093, 10.1016/S0961-9534(96)00034-7, 10.1007/BF03326184, 10.4103/2423-7752.181802, BioResources 2016 - 11 (2) – 4485 and 10.1016/j.proeng.2016.06.463).

In the journal (DOI: 10.1016/j.jclepro.2014.06.093), the researcher used the H_3PO_4 with the concentration of 85%. The high concentration of H_3PO_4 will caused the reduction of the total yield of carbon produced. The excess H_3PO_4 will promote gasification of char and increased the total weight loss of carbon [1]. The same result was also observed by other researchers [2,3]. Larger pores, which correspond to smaller surface area, develop as more H_3PO_4 are used. When pore reach a particular size in the range from mesopore to macropore, they do not contribute to the surface area significantly [4]. Apart from that, an excess of phosphorus-

containing substances in water bodies causes damage through eutrophication. Eutrophication occurs when increased levels of phosphorus lead to excessive growth of algae. The resulting "algal bloom" chokes other aquatic life, starving them of oxygen and other nutrients and blocking sunlight. Phosphates also persist for long periods in aquatic environments and are recycled back into the environment when plants decompose.

The research from (10.1016/S0961-9534(96)00034-7) has discuss the overview on quantity and potential usage biomass residues from palm oil mills in Thailand. In this manuscript the researcher only mentioned there is a possibility that the PKS can be used for activated carbon productions or charcoal. They did not discuss further about the method to convert the biomass to adsorbent material. The present work stated in my manuscript has explores the possibility of incorporating chemical pretreatment and carbonization process to create bioadsorbents from CS and PKS with high adsorption value. The effort to produce alternative high grade and inexpensive adsorbent derived from lignocellulosic biomass, particularly in the nut shell form was implied in this research.

The research from (10.1007/BF03326184 and BioResources 2016 - 11 (2) – 4485) study the sorption capacity for lead, copper and nickel and fluoride, respectively while the journal (10.4103/2423-7752.181802) and shows the methylene blue adsorption of adsorbent. The present study has focus the iodine adsorption for bioadsorbent. According to Nunes and Guerreiro [5], the iodine molecule is relatively small with an area of 0.4 nm^2 and can enter in the smaller micropores. Indeed, the methylene blue molecule has an area of 2.08 nm^2 and can only enter in large micropores and mesopores. Iodine adsorption is a simple and fast adsorption method to determine the adsorptive capacity of activated carbon. Moreover, iodine adsorption has a correlation with surface area of activated carbon. The larger the iodine number, the greater the ability of activated carbon in adsorbing the adsorbent or solute [6,7]. Iodine adsorption usually increase with increasing surface area. The adsorptive capacity of adsorbent has tight relation with its internal surface area and pore volume. The pore volume limits the size of the molecules that can be adsorbed whilst the surface area limits the amount of material which can be adsorbed, assuming a suitable molecular size. In short, iodine number indicates the development of pore [8] which are more accurate to prove the performance of an adsorbent.

The study of (10.1016/j.proeng.2016.06.463) has prepare activated carbon from palm kernel shell and coconut shell by physical steam activation and chemical activation with zinc chloride (ZnCl_2). Among the acid-based chemical agent, H_3PO_4 and ZnCl_2 are most commonly used as activation reagents. The common feature of all substances used in the chemical activation process is that they are dehydrating agents which influence pyrolytic decomposition and inhibit the formation of ash content, thus enhancing the yield of activated carbon [9]. However, using chemical agent ZnCl_2 will emitting zinc which will cause seriously environmental problems, which is strongly limits its present use [10]. Among the numerous dehydrating agents for chemical activation, the use of H_3PO_4 is preferred recently due to some environmental and profitable concerns [11]. According to Bergna et al. [13], physical activation with steam will reduces the total carbon content. Lower carbon values can be explained with the use of steam in the activation process. Steam partially oxidizes the carbon, creating a porous structure during the activation, resulting in a higher ash content but a lower carbon content.

Reviewer 2:

Comments to the Author(s)

1. There are some mistakes in the format of the manuscript. For example, the abstract is missing and the author needs to check the format of the first paragraph of all chapters.

Response: The mistakes mentioned has been corrected and clearly marked in p.1.

2. The author needs to modify the format of this manuscript according to the requirements of journals.

Response: The mistakes mentioned has been corrected and clearly marked in p.1, 2, 3, 4, 5, 9, 10, 11, 12, 13 and 14.

3. It seems that all the figures in this manuscript are drawn with “Excel”. “Origin” is a better choice. It is suggested to modify all figures in this manuscript.

Response: All the figure which drawn with “Excel” has change to “origin” as per suggestion. The figures included the Figure 4 - N₂ – adsorption isotherm in p.11, Figure 5 –FTIR in p.12 and Figure 13- TGA in p.13.

4. Please modify the unit in Table 6.

The mistakes mentioned has been corrected and clearly marked in Section 3.4, p.10.

5. Please provide more advantages about this material for its application prospect.

The advantages and references has been added and clearly marked and discussed in the manuscript. (Section 4.0. p.14)

Bioadsorbent is widely used in waste water treatment for adsorption of organic substances and non-polar adsorbates. It is the most widely used adsorbent as most of its chemical and physical properties can be tuned during the production according to its usage and preference. Moreover, the bioadsorbents derived from CS and PKS were detected to have high specific surface area due to its high porous surface and high adsorption capacity. These results show that H₃PO₄ pretreated bioadsorbent is a promising catalyst support material.

Reviewer 3

Comments to the Author(s)

1. How did the adsorption experiment conduct? Please describe it.

The adsorption experiment was added and depicted in Section 2.4, p.3.

ASTM D4607-94 method was used to determine the iodine adsorption number for carbons. The iodine adsorption number was explained as the milligrams of iodine adsorbed by 1.0 g of carbon. A conical flask with 10ml of 5% HCl and 1.0 g of activated carbon was swirled until the entire activated carbon was wetted. The wetted activated carbon was boiled for exactly 30 s and the solution was cooled to room temperature. 100ml of 0.1N (0.1 Mol L⁻¹) iodine solution was then added to the mixture of the conical flask. The mixture was later filtered using a Whatman 2V filter paper. Finally, 50ml of this filtrate was titrated with 0.1 N (0.1 Mol L⁻¹) sodium thiosulphate in the presence of starch as indicator. The amount of iodine amount adsorbed per gram of carbon was calculated as shown in equation 1:

$$\text{Iodine adsorption, } \left(\frac{\text{mg}}{\text{g}}\right) = \frac{\{(N_1 \times 126.93 \times N_2) - [(S_1 + H_1)/F_1] \times (S_1 \times 126.93) \times S_2\}}{M} \quad [\text{Equation 1}]$$

Where:

N₁ = Iodine solution normality

N₂ = Added volume of iodine solution

H₁ = Added volume of 5% HCl, ml

F₁ = Filtrate volume used in titration, ml

S₁ = Sodium thiosulfate solution normality

S₂ = Consumed volume of sodium thiosulfate solution, ml

M = Mass of carbon, g

2. To understand the adsorption process, please provide reasonable and detailed adsorbing mechanism.

The details of the adsorbing mechanism were added and depicted in Section 3.3, p.9.

Elemental iodine can be bound to activated carbon by either chemisorption or physical adsorption. Physical adsorption is the initial method of adsorption on activated carbon, which is due to its large surface area and pore structure [14].

3. How does your work compete with the existing ones towards iodine adsorption? Please provide a comparative and comprehensive explanation.

The table to highlight the efficiency of the present bioadsorbent as compared to the other biomass-derived carbon materials has been added in the manuscript (Table 6, p.8). Comparison was made on the iodine adsorption properties of the bioadsorbents from different carbonization conditions under various treatment temperature and time. The detailed discussion and reasons were added and clearly marked in p.8. References were added to provide a clear understanding for readers.

Table 6 Comparison of preparation and characteristics of bioadsorbent from this work with other studies

References	Biomass	Activation condition	Iodine adsorption(mg/g)
Present work	CS	H ₃ PO ₄ pretreatment and activate with temperature of 714°C	462.28
Present work	PKS	H ₃ PO ₄ prereatment and activate with temperature of 715°C	423.19
[15]	CS	Using temperature 700°C with activation time of 75 min	348.74
[15]	PKS	Using temperature 700°C with activation time of 75 min	304.97
[16]	Jute	Activated with N ₂ and steam with temperature 700°C for 1h	338.00
[8]	Bamboo	Using temperature 650°C with activation time of 30 min	237.00

Iodine adsorption is the most fundamental parameter used to define and characterize the performance of adsorbent. As stated by Lee *et al.* [17], aside from affecting the development of the pores, H₃PO₄ pretreated bioadsorbent might have its unique structure which would considerably affect its adsorption behaviour. Table 6 shows the characteristics of bioadsorbents produced in this work and also from other recent studies by other researchers. The table shows the results of iodine adsorption of the bioadsorbent made from different precursors and activation condition. Among the findings, bioadsorbents produced in this study apparently possesses the best quality in terms of its adsorption capability. In other words, they have higher adsorption sites for molecules to attach onto the surface of the carbon. H₃PO₄ impregnation not only promotes the pyrolytic decomposition of raw material but also leads to the formation of cross-linked structure [17,18].

References

1. Yakout SM, El-Deen GS. 2011 Characterization of activated carbon prepared by phosphoric acid activation of olive stones. *Arab J Chem*, 7.
2. Timur S, Kantarli IC, Ikizoglu E, Yanik J. 2006 Preparation of activated carbons from oreganum stalks by chemical activation. *Energ. Fuel*. **20**, 2636–2641.
3. Jagtoyen M, Derbyshire F. 1998 Activated carbons from yellow poplar and white oak by H₃PO₄ activation. *Pergamon* **36**, 1085–1097.
4. Gratuito MKB, Panyathanmaporn TP, Chunmnanklang R-A, Sirinuntawittaya N, Dutta A. 2008 Production of activated carbon from coconut shell: Optimization using response surface methodology. *Bioresour Technol* **99**, 4887–4895.
5. Nunes CA, Guerreiro MC. 2011 Estimation of surface area and pore volume of activated carbons by methylene blue and iodine numbers. *Quim Nova* **34**, 472–476. (doi:10.1590/S0100-40422011000300020)
6. Esterlita O, Herlina N. 2015 Pengaruh penambahan aktivator ZnCl₂, KOH dan H₃PO₄ dalam pembuatan karbon aktif dari pelepah aren (*Arenga Pinnata*). *J Tek Kim* **4**, 47–52.
7. Rochaeni H, Lestari PS, Tedja T, Riani E. 2018 The influence of phosphoric acid activation of carbon from Bintaro fruit (*Cerbera odollam Gaertn*) on the adsorption of chromium in various conditions of pH. *Int J Chem Stud* **6**, 443–448.
8. Mahanim SMA, Asma IW, Rafidah J, Puad E, Shaharuddin H. 2011 Production of activated carbon from industrial bamboo wastes. *J Trop For Sci 417-424* **23**, 417–424.
9. Soleimani M, Kaghazchi T. 2007 Agricultural waste conversion to activated carbon by chemical activation with phosphoric acid. *Chem Eng Technol* **30**, 649–654. (doi:10.1002/ceat.200600325)
10. Titirici M-M. 2013 *Sustainable carbon materials from hydrothermal process*. London, UK: John Wiley & Sons, Ltd.
11. Örkün Y, Karatepe N, Yavuz R. 2012 Influence of temperature and impregnation ratio of H₃PO₄ on the production of activated carbon from hazelnut shell. *Acta Phys Pol A* **121**, 277–280.
12. Guo Y, Rockstraw DA. 2007 Physicochemical properties of carbons prepared from pecan shell by phosphoric acid activation. *Bioresour Technol Vol* **98**, 1513–1521.
13. Bergna D, Varila T, Romar H, Lassi. U. 2018 Comparison of the properties of activated carbons produced in one-stage and two-stage processes. *J Carbon Res C* **4**, 41. (doi:10.3390/c4030041)
14. Jubin RT. 1979 A literature survey of methods to remove iodine from off-gas streams using solid sorbents.
15. Lee CL, H'ng PS, Paridah MT, Chin KL, Khoo PS, Nazrin RAR, Asyikin SN, Mariusz M. 2017 Effect of reaction time and temperature on the properties of carbon black made from palm kernel and coconut shell. *Asian J Sci Res* **10**, 24–33. (doi:10.3923/ajsr.2017.Research)
16. Asadullah M, Rahman MA, Motin, Mohammad Abdul Sultan MB. 2007 Adsorption studies on activated carbon derived from steam activation of jute. *J.Surface Sci.Technol* **23**, 73–80.
17. Lee CL *et al.* 2018 Production of bioadsorbent from phosphoric acid pretreated palm kernel shell and coconut shell by two-stage continuous physical activation via N₂ and air. *R Soc Open Sci* **5**, 180775. (doi:10.1098/rsos.180775)
18. Gottipati R. 2012 Preparation and characterization of microporous activated carbon from biomass and its application in the removal of chromium (VI) from aqueous Phase. National Institute of Technology Rourkela, Odisha.

We revised the whole manuscript carefully and amendment has been made on the inconsistencies/mistakes mentioned by the reviewer.